# Natural Compounds with Antiviral Activity Against Clinically Relevant RNA Viruses: Advances of the Last Decade

**DOI:** 10.3390/biom15101467

**Published:** 2025-10-16

**Authors:** David Mauricio Cañedo-Figueroa, Daniela Nahomi Calderón-Sandate, Jonathan Hernández-Castillo, Manuel Josafat Huerta-Garza, Ximena Hernández-Rodríguez, Manuel Adrián Velázquez-Cervantes, Giovanna Berenice Barrera-Aveleida, Juan Valentin Trujillo-Paez, Flor Itzel Lira-Hernández, Blanca Azucena Marquez-Reyna, Moisés León-Juárez, Ana Cristina García-Herrera, Juan Fidel Osuna-Ramos, Luis Adrián De Jesús-González

**Affiliations:** 1Laboratorio de Virología Molecular, Unidad de Investigación Biomédica de Zacatecas, Instituto Mexicano del Seguro Social, Zacatecas 98000, Mexico; davidmauricio013@gmail.com (D.M.C.-F.); 20202095@uaz.edu.mx (D.N.C.-S.); jonathan.hernandez@cinvestav.mx (J.H.-C.); huertamanuelj1d@gmail.com (M.J.H.-G.); 37183498@uaz.edu.mx (X.H.-R.); taneiro87@hotmail.com (J.V.T.-P.); flor.lihe@gmail.com (F.I.L.-H.); blanca.marquezr@uap.uaz.edu.mx (B.A.M.-R.); ana.garciaher@imss.gob.mx (A.C.G.-H.); 2Faculty of Medicine, Autonomous University of Sinaloa, Culiacan 80246, Mexico; 3Department of Infectomics and Molecular Pathogenesis, Center for Research and Advanced Studies (CINVESTAV-IPN), Mexico City 07360, Mexico; 4Departamento de Inmunología, Instituto de Investigaciones Biomédicas, Universidad Nacional Autónoma de México, Circuito Mario de La Cueva s/n, C.U, Coyoacán, Mexico City 04510, Mexico; adrianvela18@gmail.com; 5Laboratorio de Biología Molecular, Centro Nacional de Referencia II Valle de México, Salud Digna, Tlalnepantla de Baz 54075, Mexico; 6Laboratorio de Biomembranas, Departamento de Bioquímica, Escuela Nacional de Ciencias Biológicas, Instituto Politécnico Nacional, Mexico City 11340, Mexico; aveleida@gmail.com; 7Laboratorio de Virología Perinatal y Diseño Molecular de Antígenos y Biomarcadores, Departamento de Inmunobioquimica, Instituto Nacional de Perinatología “Isidro Espinosa de los Reyes”, Mexico City 11000, Mexico; moisesleoninper@gmail.com; 8Laboratorio de Virología y Diseño de Antivirales, Facultad de Medicina, Universidad Autónoma de Sinaloa (UAS), Culiacan 80246, Mexico

**Keywords:** natural products, RNA viruses, antiviral therapy, drug repurposing, bioactive compounds

## Abstract

RNA viruses remain a significant public health concern due to their rapid evolution, genetic variability, and capacity to trigger recurrent epidemics and pandemics. Over the last decade, natural products have gained attention as a valuable source of antiviral candidates, offering structural diversity, accessibility, and favorable safety profiles. This review highlights key replication mechanisms of RNA viruses and their associated therapeutic targets, including RNA-dependent RNA polymerase, viral proteases, and structural proteins mediating entry and maturation. We summarize recent advances in the identification of bioactive compounds such as flavonoids, alkaloids, terpenes, lectins, and polysaccharides that exhibit inhibitory activity against clinically relevant pathogens, including the Influenza A virus (IAV), human immunodeficiency viruses (HIV), dengue virus (DENV), Zika virus (ZIKV), and Severe acute respiratory syndrome coronavirus 2 (SARS-CoV-2). Special emphasis is placed on the integration of in silico screening, in vitro validation, and nanotechnology-based delivery systems that address challenges of stability, bioavailability, and specificity. Furthermore, the growing role of artificial intelligence, drug repurposing strategies, and curated antiviral databases is discussed as a means to accelerate therapeutic discovery. Despite persistent limitations in clinical translation and standardization, natural products represent a promising and sustainable platform for the development of next-generation antivirals against RNA viruses.

## 1. Introduction

The use of natural products offers unique and special advantages, particularly in clinical or biomedical contexts. Their low cost of acquisition, their modification, and their reduced replication cost using biotechnological techniques [1].

In the biotechnological field, various advances have been made in productive development to harness the properties of biological matrices, including plants, bacteria, fungi, marine organisms, and animals. Some specific examples are the production of antibodies and antigens from a diverse variety of microalgae [2], making affordable treatments available in regions affected by viral infections and thus globalizing the use of therapies [3].

In the search for biomolecules of interest, a key area of research is the study and discovery of treatments for diseases, specifically viral diseases. An advantage of this problem is the wide biochemical diversity of organic matrices, because within this wide diversity of products of interest, it is possible to define multiple mechanisms against various viruses, such as the use of naringenin, catechins, and keratins, which have demonstrated antiviral capacity against the SARS-CoV-2, or the use of flavonoids that inhibit the entry of a broad group of respiratory viruses [4].

It should be emphasized that one of the most interesting approaches to the development of antiviral therapies is synergy, where attempts are made to identify compounds that can enhance the efficacy of already approved treatments, such as the use of curcumin, which can inhibit viral proteases and thus modulate the immune response in respiratory diseases, establishing adjuvant drug strategies [5,6].

## 2. RNA Viruses: Biology and Mechanisms of Action

### Characteristics of RNA Viruses

RNA viruses are defined as a broad and specific group whose defining characteristic is the chemical nature of their genetic material, which is composed of ribonucleic acid (RNA). Furthermore, for their replication to be completely successful, they depend on a protein that can replicate their genetic material, RNA-dependent RNA polymerase (RdRp). This protein is encoded in the viral genetic material itself, and for it to be synthesized, the catalytic machinery of the host cell must be used when viral particles infect [7].

Several infectious diseases are caused by RNA viruses, among which the following stand out: the family *Orthomyxoviruses*, hepatitis C virus (HCV), Ebola, SARS-CoV-2, IAV, measles, as well as retroviruses such as human T-cell lymphotropic virus (HTLV-1) and HIV. In this sense, RNA viruses can be classified into viruses with single-stranded RNA in both positive (+ssRNA) and negative (−ssRNA) senses, ambisense RNA, and double-stranded RNA (dsRNA). These viral agents can regulate the presence of RdRPs for replication of their genomes, or, in retroviruses, with two copies of single-stranded RNA genomes, reverse transcriptase (RT) produces viral DNA that can integrate into host DNA under its integrase function [8].

In 1971, Baltimore proposed a classification system for viruses based on the type of genetic material and replication strategy, emphasizing the production of messenger RNA (mRNA) [9,10]. This classification is summarized in Table 1, which focuses solely on the classification of RNA viruses.

## 3. The Secret to Therapeutic Targets: Replication Mechanisms

Replication mechanisms are fundamental because they dictate, at the molecular level, the relationship between viruses and the cells they infect. Understanding these processes provides the basis for identifying conserved therapeutic targets, many of which are common across diverse RNA viruses [16].

Among them, RdRp stands out as the primary target due to its essential role in genome replication, its high conservation, and the lack of homologous proteins in human cells, which increases specificity and reduces toxicity [17,18]. Structurally, RdRp contains catalytic domains with conserved motifs that coordinate RNA synthesis [19].

Another key group is the viral proteases, which process large polyprotein precursors into functional proteins. Coronaviruses encode Papain-Like protease (PLpro) and main protease (Mpro), both essential for replication [20,21]. Flaviviruses depend on the NS3 serine protease [22,23,24,25,26,27,28], while picornaviruses encode the 3C cysteine protease [29]. These proteases share functional conservation, regulating replication phases and immune evasion [30,31,32]. Importantly, inhibitors such as nirmatrelvir exemplify how protease conservation enables broad-spectrum antiviral development [33].

RNA viruses have also evolved immune evasion strategies, including the compartmentalization of RNA in double-membrane vesicles and the inhibition of RIG-I/MDA5 signaling. Compounds such as K22 disrupt RNA compartmentalization [34], while accessory proteins in paramyxoviruses block MDA5 detection of dsRNA [35,36]. In contrast, agonists of RIG-I or MDA5, or synthetic triphosphorylated RNAs, have been explored as immunomodulatory antivirals [37].

Finally, retroviruses (e.g., HIV, HTLV) employ RT to convert RNA into DNA and integrate into the host genome [38,39]. Given their dependence on RT, this enzyme is another primary therapeutic target for designing broad-spectrum antivirals [40].

In summary, conserved viral enzymes such as RdRp, proteases, and RT, along with host immune-sensing pathways, represent central therapeutic targets that can be exploited by natural compounds, as will be discussed in the following sections.

## 4. Materials and Methods

To ensure a comprehensive and unbiased overview, we conducted a structured literature search of studies published between January 2013 and June 2024. The databases PubMed, Scopus, and Web of Science were queried using combinations of the following keywords: “*RNA viruses*,” “*antiviral activity*,” “*natural compounds*,” “*flavonoids*,” “*alkaloids*,” “*terpenes*,” “*lectins*,” “*polysaccharides*,” and “*nanodelivery*”. Only peer-reviewed original articles and reviews reporting experimental data (in vitro, in vivo, or clinical studies) were considered. Reports exclusively based on in silico approaches were included when they provided novel insights into molecular interactions or guided experimental validation.

We prioritized studies describing clinically relevant RNA viruses such as IAV, DENV, ZIKV, HCV, HIV, and coronaviruses (including SARS-CoV and SARS-CoV-2). Additional viruses were included when sufficient mechanistic or pharmacological evidence was available. Articles were excluded if they lacked information on biological activity, pharmacological parameters (e.g., cytotoxic concentration 50% (CC_50_), and effective concentration 50% (EC_50_)), or did not specify the natural origin of the compound.

## 5. Natural Antivirals: Ten Years Ago

The use of molecules obtained from natural matrices offers significant advantages, as mentioned in the introduction to this work (Figure 1).

Nevertheless, a comprehensive and up-to-date review of the most relevant advances in antiviral therapy is necessary, considering the primary therapeutic approaches against these pathogenic agents. RNA viruses require adhesion to the cell wall, primarily through endocytosis or membrane fusion, where certain compounds are being investigated to prevent the infection from worsening in the early stages of the disease.

### 5.1. Inhibition of Replication

Genetic material replication is one of the most studied areas of interest in the development of antiviral therapies, due to its crucial role in conserving viral genetic material. Strategies targeting proteins such as RdRp or RT have been proposed as a first approach to highly effective antiviral therapies [41].

The search for antivirals has led to the development of numerous strategies to generate approaches for obtaining beneficial compounds, including in silico simulations to filter or identify possible interactions with proteins involved in genetic material replication. An extensive in silico evaluation assessed 1664 substances approved by the FDA versus SARS-CoV-2 RdRp, reporting eight natural compounds with high binding scores (asiaticoside, glycyrrhizin, aloin, digoxin, sennoside B, quercetin, taxifolin, and neohesperidin dihydrochalcone), which was a watershed moment for characterization and evaluations of SARS-CoV-2 replication [42].

Other proteins associated with viral replication independent of RdRp have also been studied. Compounds such as epigallocatechin gallate (EGCG) have been reported to inhibit NSP15 at inhibitory concentration 50% (IC_50_) values of 2.54 μg/mL, in addition to inhibiting SARS-CoV-2 replication in Vero cells at concentrations of 0.24 μg/mL [43].

Calonalidine A, present in Malaysian *Calophyllum* spp. trees, show potent inhibition against HIV-1 RT in vitro assays, as well as being transportable to preliminary clinical trials, where, in healthy volunteers, it showed an acceptable safety profile [44].

Other potentially practical examples for RT inhibition are the peptides AIHIILI and LIAVSTNIIFIVV isolated from acorn husks (*Quercus infectoria*). These peptides showed IC50 of 274 and 236.3 nmol, respectively [45].

### 5.2. Inhibition of Host Cell Entry

Viral particles bind to the cell membrane through surface protein structures that recognize each other and allow the particles to bind together for subsequent introduction into the intracellular space [46].

One of the most developed antiviral strategies is the use of proteins that have an affinity for viral carbohydrates, such as plant lectins. Among these, BanLec stands out, a lectin obtained from bananas that has an affinity for binding to mannose residues of the gp120 glycoprotein of HIV [47], thereby preventing binding to CD4 receptors [48]. BanLec was engineered with point mutations and renamed H48T. When tested in HIV-infected mice, it showed the same efficiency as in vitro studies [49].

One strategy employed by a considerable number of viruses is to prevent these proteins from interacting with each other, thereby halting subsequent viral steps. Coronaviruses have the spike (S) protein on their surface, which is recognized by multiple receptors, thus allowing entry [50]. It is therefore logical to consider strategies that inhibit this interaction.

Lectins are a diverse group of non-enzymatic proteins, many of which are plant-derived, and have been studied for their applications in combating infectious diseases caused by viruses, such as HIV [51]. Among these, BanLec, a lectin obtained from bananas, has been investigated as an antiviral agent in infections caused by HIV-1 and HIV-2. Its mechanism of action has been determined as the binding of the lectin to mannose-type glycans present in the gp120 glycoprotein of the viral envelope, subsequently blocking the host cell receptor CD4, as well as the CCR5 and CXCR4 receptors [52].

Algae also represent a promising source of natural compounds for use against infectious diseases. The red marine alga *Griffithsia* sp. has been studied for the presence of griffithsin, a lectin capable of inhibiting certain coronaviruses, such as MERS-CoV, in MRC-5, Vero, and Huh-7 cells. Its mechanism of action has been established as the inhibition of angiotensin-converting enzyme 2 (ACE2) [53].

Pentacyclic terpenoids are secondary metabolites resulting from the biological processes of some plants. These hydrophobic compounds encompass a wide variety of structures. Acids such as oleanolic and echinocystic are pentacyclic groups of terpenoids that, in HCV infections in Huh-7 cells, have demonstrated antiviral activity by blocking viral envelope glycoproteins [53].

Quercetin, a flavonoid extracted from a broad range of plants, has been characterized for decades as a potential antiviral agent against SARS-CoV-2, particularly in combination with vitamin C [54]. These findings provide sufficient evidence to extrapolate the study to other viruses to elucidate a possible mechanism of action. In IAV infections, quercetin has been shown to interact with the HA2 protein, inhibiting replication by interfering with hemolysis or endocytosis, suggesting a possible interaction with the viral fusion membrane [55].

Other studies aimed at elucidating the mechanism of action against SARS-CoV-2, as well as determining whether other flavonoids could be effective, found that isorhamnetin significantly reduced the entry of a pseudotyped virus in HEK293 cells overexpressing ACE2. However, quercetin did not appear to interact with ACE2 in the same manner as isorhamnetin, suggesting that quercetin is unlikely to be effective against SARS-CoV-2 infections [56].

Other biomolecules, such as lipid extracts, also known as essential oils, are being studied due to their widespread presence in a diverse range of plant species. Investigations have reported antiviral activity mainly from extracts obtained from *Citrus × aurantifolia* and *Geranium phaeum*. These extracts have been reported to act as ACE2 inhibitors and to reduce the expression of this protein in HT-29 cells, suggesting not only genetic regulation but also a potential blocking effect against infections targeting this receptor [57].

Strategies such as virtual screening enable the identification of numerous compounds that could interact with one or more targets in i*n silico* simulations. Approximately 78 compounds of natural origin have been reported against the three main SARS-CoV-2 proteins (main protease, spike, and RdRP), yielding diverse results but providing a solid basis for in vitro evaluations [58].

Protein compounds present in *Curcuma longa*, known as CI, CII, and CIII, showed strong affinities and stability with hemagglutinin, a protein associated with the binding of various IAV [59,60].

### 5.3. Protein Processing Inhibitors

The late stages of infection begin once viral particles manage to enter the host. However, because symptoms can be mild, it is possible to initiate treatments in the early stages of infection, which can be helpful.

Components like alkaloids have been reported as a valuable strategy to stop the viral machinery from assembling, especially berberine, which comes from *Berberis vulgaris* and *Coptis* sp. This, in mice and cell lines like A549, LET7, and HAE infected with IAV, prevented the nuclear export of viral ribonucleoproteins to the cell nucleus by inhibiting mitogen-activated protein kinase/extracellular signal-regulated kinase 1 (ERK/MAPK), induced at the onset of viral infection [61].

Zeylanone epoxide, a biomolecule obtained from *Diospyros anisandra*, has been reported to have antiviral activity, inhibiting two key stages of the IAV and *Influenzavirus* B (IBV) virus replication cycle, specifically at the entry and middle stages. A potential mechanism of action of zeylanone was first evaluated by assessing the presence of nucleoproteins when confined to the cell nucleus, indicating that this biomolecule affects intracellular distribution and reduces viral yield [62].

Luteolin, a natural flavonoid found in plants such as celery, carrots, and broccoli, was used as a treatment to block DENV maturation in Huh-7 cells infected with dengue serotypes, allowing for the determination of EC_50_ and CC_50_ values. When analyzing the recovered virions, high levels of precursor membrane protein (prM) were determined after treatment with luteolin, suggesting the immaturity of these viral particles. PrM, together with the E protein, allows for the encapsulation and protection of immature virions, which are subsequently cleaved by furin [63].

The efficacy of luteolin was determined in murine models, showing that the viral titer in blood samples was reduced by half compared to the control used. The authors conclude that this low efficiency may be due to the limited bioavailability of the treatment or its low potency in higher organisms. They suggest strategies to optimize these compounds or the administration of the drug [64].

DENV protease NS2B-NS3 is considered one of the most common targets in this infection. Inhibitors such as agathisflovana, a biflavonoid obtained from *Poincianella pyramidalis* (Tul.), have been evaluated for their effect. The results of in vitro studies have found that this protease is inhibited at an IC50 of 15 μM. These results were confirmed by bioinformatics techniques, which confirmed that this treatment binds to the catalytic site of NS2B-NS3 [65].

Using nuclear magnetic resonance and crystallography techniques, it has been determined that curcumin, a polyphenol obtained from *Curcuma longa*, exhibits allosteric activity toward the NS2B-NS3 protease cavity, stabilizing it toward an inactive conformation [66]. However, although curcumin has been reported as a weak inhibitor in vitro evaluation, trials have been conducted in which curcumin derivatives improved stability and antiviral activity in paca assays against DENV2 as a skeleton molecule for future, more effective treatments [67].

Terpenes obtained from sources such as *Azadirachta indica* have been shown to exhibit affinity in simulated models against DENV protease, forming hydrogen bonds at crucial catalytic sites such as His51, Asp75, and Ser135 [68]. There are similar reports in evaluations of biomolecules from medicinal fungi, *Ganoderma lucidum*, which showed inhibition against dengue protease in silico, and later in in vitro evaluations, demonstrating that ganodermanontriol reduced viral titer by up to 40% at concentrations of 50 μM [69].

Earlier studies demonstrated the effectiveness of betulinic acid, a naturally occurring pentacyclic terpenoid found in plants such as *Eucalyptus* spp., against HIV, elucidating its mechanism by blocking the cleavage of the mature p24 capsid protein. However, clinical trials, which reached phase II, halted progress when they found resistance to the drug. Therefore, it has been determined that botulinic acid derivatives have been able to be the first treatment to inhibit HIV-1 maturation by preventing the gag polyprotein from undergoing final cleavage, where the mature capsid p24 separates into the precursor p25 [70].

Compounds such as theaflavins, a group of polyphenols found in oolong teas like *Camellia sinensis*, particularly theaflavin-3-3′-digallate and 3-isotheaflavin-3 gallate, can inhibit the Mpro of the SARS-CoV-2, as well as inhibit Nsp12, one of the RdRp in those viruses [58].

Taking advantage of computational power, virtual approximations of compounds obtained from natural sources, such as silymarin, a polyphenol present in artichokes; palmatine, an alkaloid present in plants like *Coptis chinensis*; and curcumin, a diferuloylmethane present in *Curcuma longa*, were performed. All three compounds showed affinity and stability compared to drugs already used for viral diseases, such as hydroxychloroquine and remdesivir [71].

### 5.4. Favorable Immune Compounds

Compounds reported as micronutrients are essential for proper cellular and biological function. They have been relevant in clinical studies, as in cases of infections with a high risk of mortality, such as measles, pneumonia, or diarrheal diseases, these compounds are absent or present in low concentrations [72].

Molecules not classified as vitamins or minerals can also modify the immune response. Plant species such as *Astragalus membranaceus* are being studied for their active compound astragaloside IV (AS-IV), a triterpenoid compound with broad anti-inflammatory, antibiotic, and antiviral properties [73]. In vitro studies using A549 reported how AS-IV regulates the production of reactive oxygen species (ROS) in cultures infected with IAV, thereby inhibiting molecules such as inflammasomes and caspase-1, which reduces the levels of interleukin (IL)-1β and IL-19 [74].

Alkaloids such as Berberine have been presented as an antiviral strategy, suppressing TLR7 upregulation signaling, such as MyD88 and NF-kB (p65), as well as significantly inhibiting the increase in Th1/Th2 and Th17/Treg ratios, as well as inflammatory cytokines induced by viral infection in mice with viral influenza [75].

*Houttuynia cordata*, an Oriental herb traditionally used to treat pneumonia, has been shown to contain flavonoids and polysaccharides. In mice infected with the IAV subtype H1N1, the combination of flavonoids and polysaccharides increased mouse survival and improved lung parameters [76].

Alterations in extraction processes also influence the results of new compounds. The use of ethanol in *Houttuynia cordata* yielded extracts that, in A549 and alveolar macrophages (MH-S), served to reduce inflammatory markers such as IL-6 and NO, showing interesting results in murine models with the administration of flavonoids at 100 and 400 mg/kg, reducing inflammation by 46.1 and 66.5%, respectively [77].

Glycyrrhizin, a type of triterpenoid saponin obtained from licorice root (*Glycyrrhiza glabra*) [78], has been reported to have various antiviral properties, such as increasing nitric oxide production in macrophages, altering cell signaling pathways, or transcription factors such as AP-1 and NF-κB, as well as interacting with membrane receptors such as ACE2 in Vero cells [79]. When combined with a nitric oxide donor such as BETA NONOate under culture conditions, this synergy inhibits SARS-CoV-2 viral replication [80].

Saponoids, alkylamides, and polysaccharides are some of the compounds not classified as vitamins or minerals that have demonstrated immunological regulations of interest. Experiments have shown that extracts obtained from the root and flower tips of *Echinacea purpurea* have been shown to increase the phagocytic capacity of natural killer cells and the production of T lymphocytes [81].

Bushes native to China, like *Camellia sinensis L.*, have also been shown to be sources for the extraction of saponins, polyphenols, and acids such as EGCG. These bushes have been studied in the context of viral infections, since it has been shown that their compounds, such as EGCC, have immunoprotective effects by inhibiting neuraminidase in IAV infections, or viral RNA synthesis in infections such as DENV, ZIKV, and *Japanese encephalitis virus* (JEV), in addition to protecting host cells from other infections by interacting with the membrane in HIV and HCV infections [82].

Although we are familiar with many biomolecules, some others remain poorly defined, and plant extracts like *Sambucus nigra* L. have been investigated in alternative medicine treatments. Belonging to the Aoxaceae family, native to Asia, Europe, and North Africa, where 80 people with influenza symptoms received a syrup extracted from *S. nigra*, showing symptoms of improvement after 3–4 days compared to the placebo group, it is hypothesized that this antiviral effect is due to the lecithins present, particularly SAα2, 6Gals, and Neu5Acα, which play an essential role in the release of cytokines such as IL-6, IL-8, and TNF-α during IV infection [83,84].

Systematic reviews have provided relevant information on immunoregulators and antivirals found in medicinal plants, identifying approximately 25 species, including *Eucalyptus globulus*, *Aloe vera*, and *Camellia sinensis*, which have positive effects on the host’s defense against viral infections such as IV, SARS-CoV-2, herpes simplex virus, or DENV, by regulating immune cells, although the mechanism of action is not reported in all cases [85].

To facilitate comparison among the different bioactive compounds reported, we summarized the main findings in Table 2, which compiles natural products with antiviral activity against RNA viruses described in the last decade. The table highlights the biological source, type of compound, experimental model, and key pharmacological values (CC_50_ and EC_50_) when available, thus providing a concise overview of their therapeutic potential.

## 6. New Challenges

### 6.1. Nanodelivery of Natural Antivirals

Many natural compounds possess characteristics that may compromise their therapeutic utility. For instance, the degradation of peptides by enzymes present in biological systems is a significant limitation that must be considered when proposing these compounds as antiviral agents. Although various strategies have been developed to overcome this drawback, nanotechnology has emerged as a key tool to address such challenges. Nanoformulations can enhance the solubility of compounds, extend their half-life, reduce systemic toxicity, protect them from premature degradation, and enable controlled release [51,86,87].

Particularly in the case of respiratory diseases such as COVID-19, where the virus is transmitted through airborne routes and replicates in the nasal and pulmonary epithelium, local delivery systems such as aerosols or nebulizers can be especially advantageous in reducing viral load and preventing respiratory complications [51,86,87].

Flavonoids have attracted attention due to their broad-spectrum antiviral activity. Recent studies have demonstrated that flavonoid-based nanoparticles can enhance stability, improve targeted delivery, and increase therapeutic efficacy. For example, nanoparticle formulations of quercetin have shown increased antiviral potency and bioavailability [88]. Reviews published in 2024 highlight that flavonoid-loaded nanocarriers (e.g., polymeric nanoparticles, liposomes, and solid lipid nanoparticles) significantly improve pharmacokinetics and therapeutic outcomes, positioning them as promising candidates against viral infections [89,90].

Similarly, terpenoids encapsulated in nanocarriers are emerging as attractive antiviral agents. A 2024 review emphasized their potential against central viral infections, underscoring how nanodelivery systems enhance antiviral activity, reduce toxicity, and improve targeting [91].

The main delivery challenges for natural antivirals (flavonoids, terpenes, peptides, lectins, and polysaccharides) can be categorized into physicochemical and biological–immunological barriers (Figure 2).

### 6.2. Natural Compounds as Nanocarriers or Antivirals for Nanodelivery

The successful use of nanotechnology has also been documented in other viral infections, including the following:HIV-1: Dendrimers, solid lipid nanoparticles (SLNs), and chitosan nanoparticles loaded with antiretroviral drugs increase bioavailability while reducing toxicity [92,93].HBV: micelles of chito-oligosaccharides and derivatives enhance encapsulation efficiency and sustained release of lamivudine, and cell-penetrating peptide (CPP)-based approaches have demonstrated inhibition of *Avihepadnavirus anatigruidae* (DHBV) and HBV replication in cell and animal models [92,94].

The development of intelligent, targeted delivery systems is emerging as one of the most promising avenues for treatment with natural antivirals. Nonetheless, technical and regulatory barriers remain, including the need to standardize manufacturing processes (e.g., PDNVs), ensure batch-to-batch reproducibility, and comprehensively evaluate pharmacokinetics, biodistribution, and long-term toxicity [95,96,97,98].

In addition to innovative nanomaterials, oral delivery strategies are also gaining relevance, not only for patient convenience but for their prophylactic utility in high-risk populations. New approaches aim to design oral antiviral materials capable of sustained release in the oral cavity, leveraging the expression of viral receptors such as ACE2 in the oral epithelium. The combination of compounds such as glycyrrhizin, dapivirine, or griffithsin with targeted-release nanoparticles shows encouraging in vitro results against SARS-CoV-2 [99].

Artificial intelligence could optimize platform design, anticipate drug–nanocarrier interactions, and guide the selection of the most suitable systems for each peptide type. Likewise, the design of hybrid nanomaterials (e.g., polymer–lipid) and nanorobots opens new frontiers in precision antiviral medicine [100]. In the specific case of peptides derived from algae, their integration into biocompatible delivery systems, such as chitosan nanoparticles, plant-derived vesicles, or marine polysaccharide matrices, represents an attractive and sustainable alternative aligned with the growing interest in natural therapies against emerging viruses such as SARS-CoV-2 [101].

## 7. Conclusions

The last decade has been pivotal in generating evidence that supports the development of new antiviral therapeutic strategies. Although each virus displays unique characteristics, proteins, and specific biological responses, the most common steps in the viral infection cycle have enabled the design of treatments applicable across diverse infectious contexts.

Extensive evidence supports the evaluation of natural molecules, such as lectins, saponins, flavonoids, and other bioactive compounds, which often exhibit specific effects. In addition, emerging technologies, including deep learning algorithms and large-scale international databases, have introduced new opportunities to accelerate the discovery of antiviral agents, considerably reducing the time required to identify promising candidates.

The rise of in silico simulations and interaction-prediction methods has accelerated and deepened the exploration of putative mechanisms of action that are expected to be reproduced in vitro. These tools enable more complex and specific understanding, support the rapid down-selection of large numbers of compounds, and help define unique therapeutic targets with the potential to be applicable across a range of viral species [48].

These new strategies help reduce costs, democratize information, and establish standardized protocols, leading to more efficient and precision-oriented workflows. They also enable international pipelines in which collaboration among diverse research groups is facilitated by extensive databases cataloging previously reported antivirals across multiple categories [102].

Among the bioinformatics servers and tools used to identify molecules with antiviral activity are the A Database of Antiviral Peptides (AVPdb), the Antiviral Drugs Resource and Machine Learning Platform (VDDB), and the Data Repository of Antiviral Peptides and Proteins (DRAVP), each with its specific characteristics. AVPdb was created to store sequences of peptides with antiviral activity against different viruses, such as HIV, RSV, or SARS-CoV-2, providing details such as the evaluated cell lines and physicochemical properties [103]. VDDB, in turn, emphasizes the management of this rapidly growing volume of data, as it is designed to predict antiviral agents using machine learning models such as Random Forest or Extreme Gradient Boosting (XGBoost) [104]. Finally, although DRAVP shares a similar approach to VDDB, this database incorporates manual curation, hosting annotations of various antiviral peptides and proteins, including registered patents, physicochemical data, references, and clinical associations [105]

Nevertheless, critical gaps remain, including the limited standardization of natural products, the insufficient in vivo validation compared to in vitro studies, and the challenges of bioavailability, since these compounds are often prone to degradation by endogenous proteins and enzymes within living systems.

This review aims to provide researchers with an updated overview of advances in the field of natural antivirals, highlighting innovative studies and strategies that can enhance research efficiency and save time while also addressing key challenges to encourage research teams to propose creative solutions that sustain and advance the development of these therapeutic approaches.

## Figures and Tables

**Figure 1 biomolecules-15-01467-f001:**
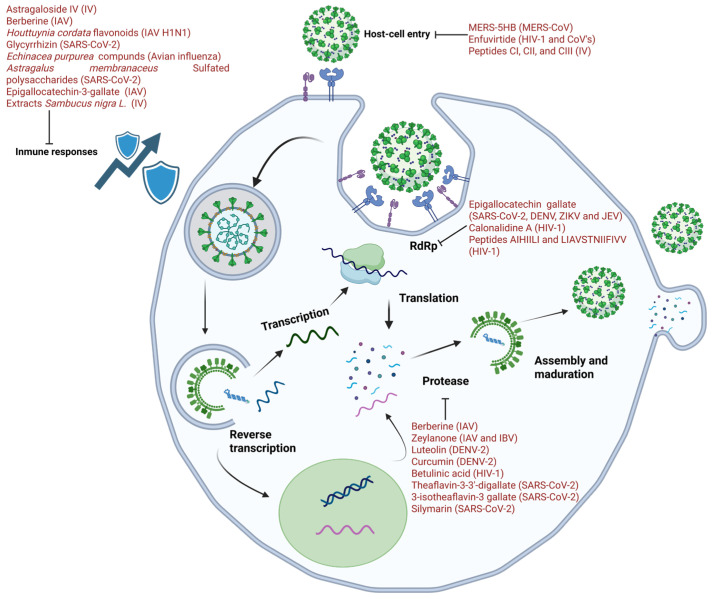
**Natural compounds targeting RNA viruses.** Bioactive molecules act at different stages of the viral cycle: **entry** (lectins, peptides, and flavonoids), **replication/RT** (EGCG, calanolide A, and acorn peptides), **protease processing** (curcumin, agathisflavone, theaflavins, and berberine), **maturation** (betulinic acid, and luteolin), and **immune modulation** (astragaloside IV, glycyrrhizin, complete extracts of *Echinacea*, and *Sambucus*).

**Figure 2 biomolecules-15-01467-f002:**
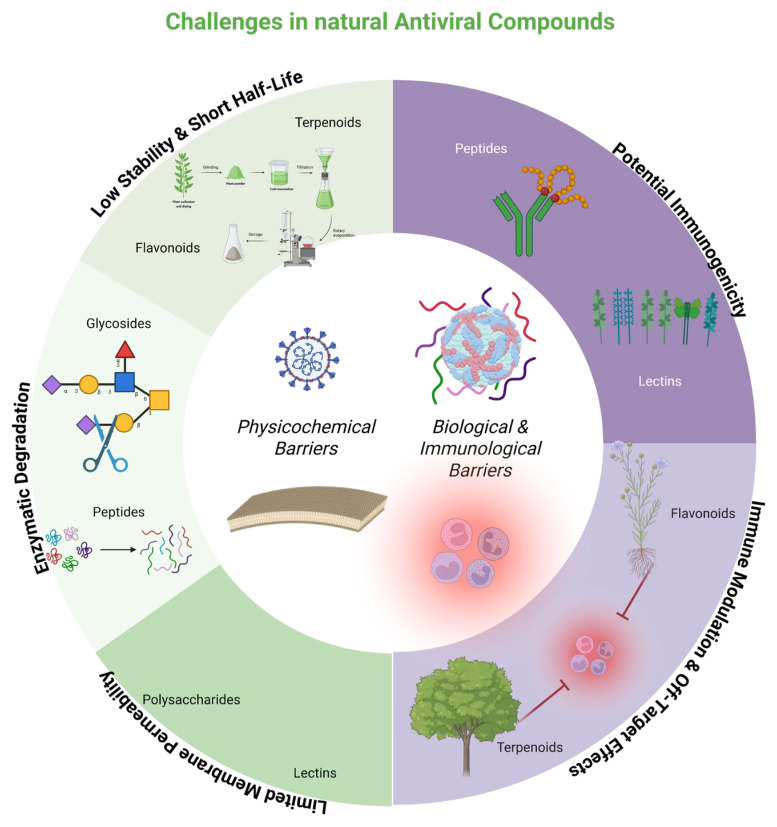
Main administration challenges of natural antiviral compounds, including peptides, flavonoids, terpenoids, polysaccharides, and lectins. Barriers can be categorized into two main types: physicochemical (e.g., enzymatic degradation, low stability, and limited membrane permeability) and biological/immunological (e.g., potential immunogenicity and off-target immune modulation). These factors collectively limit bioavailability, stability, and therapeutic efficacy, highlighting the need for nanodelivery and natural carrier platforms.

**Table 1 biomolecules-15-01467-t001:** Classification of RNA viruses by Baltimore in 1971.

Group	Virus Type	Examples	Replication Mechanism	Features of Interest	Reference
III	dsRNA	Rotavirus	Replication occurs in the cytoplasm, where RdRp transcribes the negative strand to generate functional mRNA.	Because these viruses elicit strong innate immune responses, they are key models for studying antiviral immunity.	[11]
IV	+ssRNA	ZIKV, DENV, and SARS-CoV-2	The genome can function as mRNA, allowing viral proteins to be translated quickly after entering the cytoplasm.	The viral RdRp synthesizes a −ssRNA intermediate that serves as a template for replication. Its high mutation rate and adaptability pose significant challenges for the development of antivirals.	[12]
V	−ssRNA	*Lyssavirus rabies*, IAV, and respiratory syncytial virus (RSV).	The genome cannot be translated directly; it must first be converted into mRNA by the RdRp before protein synthesis occurs.	Many viruses have segmented genomes, which promotes genetic reassortment and increases antigenic diversity.	[13]
VI	ssRNA retrovirus	HIV	They carry +ssRNA but do not use it directly as mRNA; instead, RT converts the RNA into DNA, which integrates into the host genome.	This replication strategy enables persistent infection and presents therapeutic challenges, while also providing targets for RT-directed antivirals.	[14]
VII	DsDNA retrovirus	Hepatitis B virus (HBV)	They use a pregenomic RNA intermediate during replication, which is subsequently reverse transcribed into DNA by a viral RT.	Their hybrid mechanism places them as a distinct category within the Baltimore system and raises therapeutic challenges like those of retroviruses.	[15]

dsRNA = double-stranded RNA; ssRNA = single-stranded RNA; +ssRNA = positive-sense single-stranded RNA; −ssRNA = negative-sense single-stranded RNA; RT = reverse transcriptase; RdRp = RNA-dependent RNA polymerase.

**Table 2 biomolecules-15-01467-t002:** Natural compounds with antiviral activity against RNA viruses (last decade).

Virus/Family	Compound/Extract	Biological Source	Type of Compound/Extract	CC_50_ (µM/µg/mL)	EC_50_ (µM/µg/mL)	Model (in vitro/in vivo)	Reference
SARS-CoV-2 (*Coronaviridae*)	EGCG	*Camellia sinensis* (green tea)	Flavonoid	n.d.	IC_5050_ = 2.54 µg/mL; replication inhibition at 0.24 µg/mL	Vero cells (in vitro)	[43]
	Quercetin + Vitamin C	Various plants	Flavonoid + vitamin	n.d.	Synergistic inhibition	In vitro; proposed clinical synergy	[54]
	Isorhamnetin	Various plants	Flavonoid	n.d.	Reduces pseudotyped virus entry	In vitro (HEK293/ACE2)	[57]
	Theaflavins (TF3DG, TF3G)	*Camellia sinensis* (tea)	Polyphenols	n.d.	Inhibit Mpro and RdRp (Nsp12)	In vitro (enzymatic/cell assays)	[58]
	Glycyrrhizin	*Glycyrrhiza glabra* (licorice)	Triterpenoid saponin	n.d.	ACE2 interaction; replication inhibition	In vitro (Vero cells)	[78,79,80]
MERS-CoV (*Coronaviridae*)	Griffithsin lectin	*Griffithsia* sp. (red algae)	Lectin	n.d.	Inhibition of entry	In vitro (MRC-5, Vero, Huh-7)	[53]
HIV-1 (*Retroviridae*)	Calanolide A	*Calophyllum* spp.	Coumarin derivative	n.d.	Active in vitro; safe in phase I	In vivo/human volunteers (safety)	[44]
	AIHIILI & LIAVSTNIIFIVV peptides	*Quercus infectoria* (acorn husks)	Peptides	n.d.	IC_50_ = 274 nM/236.3 nM	In vitro (RT inhibition)	[45]
	BanLec lectin (mutant H84T)	*Musa* spp. (banana)	Lectin	n.d.	Potent inhibition in vitro/in vivo	In vitro; in vivo (mice)	[47,48,49,52]
	Betulinic acid derivatives	*Eucalyptus* spp.	Pentacyclic triterpenoid	n.d.	Inhibits Gag maturation (capsid p24)	In vitro; Phase II halted	[70]
HCV (*Flaviviridae*)	Oleanolic & echinocystic acids	Plants (pentacyclic triterpenoids)	Terpenoids	n.d.	Block viral envelope glycoproteins	In vitro (Huh-7)	[53]
DENV (*Flaviviridae*)	Luteolin	Vegetables (celery, carrots, broccoli)	Flavonoid	CC_50_ reported (n.d., exact value)	EC_50_ reported (n.d., exact value)	In vitro (Huh-7); in vivo (mice, ↓ viremia)	[63,64]
	Agathisflavone	*Poincianella pyramidalis* (Tul.)	Biflavonoid	n.d.	IC_50_ = 15 µM	In vitro (NS2B-NS3 protease inhibition)	[65]
	Curcumin	*Curcuma longa*	Polyphenol	n.d.	Weak inhibition; derivative ↑ activity	In vitro (DENV2 NS3 protease); in silico	[66,67]
	Ganodermanontriol	*Ganoderma lucidum* (fungus)	Triterpenoid	n.d.	↓ viral titre by ~40% at 50 µM	In vitro	[69]
Influenza A/B (*Orthomyxoviridae*)	Zeylanone epoxide	*Diospyros anisandra*	Quinone derivative	n.d.	Entry & mid-stage replication inhibition	In vitro (IAV, IBV)	[62]
Influenza A (*Orthomyxoviridae*)	Quercetin	Multiple plants	Flavonoid	n.d.	Entry inhibition via HA2 interaction	In vitro (IAV)	[55]
	Berberine	*Berberis vulgaris; Coptis* sp.	Alkaloid	n.d.	Inhibits ERK/MAPK ↓ replication	In vitro; in vivo (mice)	[61,75]
	Astragaloside IV	*Astragalus membranaceus*	Triterpenoid saponin	n.d.	Anti-inflammatory; ↓ viral titers	In vitro (A549); in vivo (mice)	[73,74]
	Flavonoids + polysaccharides (extract)	*Houttuynia cordata*	Mixed extract	n.d.	↑ survival; ↓ lung inflammation	In vivo (mice)	[76,77]
	Echinacea extract	*Echinacea purpurea*	Extract (saponins, alkylamides, polysaccharides)	n.d.	↑ NK cell activity; T lymphocyte production	In vivo/immunomodulatory	[81]
	EGCG (broad review)	*Camellia sinensis*	Flavonoid	n.d.	Inhibits neuraminidase; viral RNA synthesis	In vitro	[82]
	Elderberry extract	*Sambucus nigra* (berries)	Extract (lectins, polyphenols)	n.d.	Symptom improvement in a clinical trial	In vivo (patients)	[83,84]

Abbreviations: n.d. = not determined; EC_50_/CC_50_ are reported when available in the cited studies.

## Data Availability

No new data were created or analyzed in this study.

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
