# Peer review of "Natural Compounds with Antiviral Activity Against Clinically Relevant RNA Viruses: Advances of the Last Decade"

_biomolecules, 2025, doi:10.3390/biom15101467_

Round 1
Reviewer 1 Report
Comments and Suggestions for Authors
The manuscript addresses a highly relevant and timely topic: the use of natural products in clinical and biomedical contexts for the treatment of RNA virus infections. The proposal is interesting and has great potential impact. However, I have identified several aspects that, in my opinion, should be revised to strengthen the clarity, focus, and coherence of the work. Below, I detail my observations with the intention of contributing to improving the quality and alignment of the manuscript with its stated objectives:
The title promises a focus on RNA viruses and natural compounds reported in the last decade. However, in some passages, it is not entirely clear what the specific role of natural compounds is in relation to the “challenge” posed by RNA viruses.
The introduction mentions the use of biological matrices as bioreactors, but this topic is not revisited in the rest of the text. I suggest either developing it with concrete background examples or removing it.
In the main body of the text, several sections provide very extensive descriptions of basic viral biology, sometimes in excessive detail (e.g., replication of influenza, rotavirus, Lassa). This may dilute the focus on natural antiviral compounds, which is what is expected according to the title and abstract. While the review is logically structured, there is an imbalance in length:
- The sections on RNA virus biology and basic replication mechanisms (chapters 2 and 3) are too extensive and detailed, with technical descriptions of well-known mechanisms that are not strictly necessary for the topic of the review. In my view, the information provided in Table 1 is sufficient. A caption should be added to Table 1 defining the abbreviations used for the viral types.
- Chapter 4, dedicated to natural compounds, is shorter and more general than expected. It would be important to expand this section with concrete examples, mechanisms of action, results from animal models, and, if possible, clinical trial data. A critical comparison of results would significantly enrich the section. A much broader search of natural compounds with antiviral activity and the viral models in which they were tested is needed (for example, doi: 10.1002/cbdv.202401913; doi: 10.3390/nu12092534; doi: 10.1016/j.heliyon.2024.e27533).
- L357: “The use of molecules obtained from organic matrices …” – what is meant by “organic matrices”? Should this not be “natural matrices”?
- L370–376: Move the entire paragraph to section 4.2, Inhibition of host-cell entry. I recommend reorganizing Chapter 4 according to the different stages of the viral life cycle (entry, replication, release) to provide greater coherence.
- Table 2 includes examples of synthetic rather than natural origin: celgosivir is a prodrug of a natural product (castanospermine) designed for oral administration and rapid conversion into its active form. UV-4B is a synthetic analog of natural DNJ, developed to improve its pharmacological and antiviral properties.
- Section 4.4 should include only studies evaluating immunomodulatory effects in the context of viral infection, not those focusing on cancer, autoimmune diseases, non-viral infections, etc.
- The entire chapter should be reorganized and rewritten with consistency regarding which data will be provided for each example (CC50, EC50, in vitro or in vivo, biological source of the sample, type of extract, type of compound involved in the reported biological activity, etc.).
- Chapter 5, focused on nanotechnology, is up to date but mostly describes applications related to synthetic antivirals. For a manuscript whose stated central focus is “natural antivirals,” this chapter appears weakly connected. I suggest emphasizing the link to natural compounds (e.g., flavonoids, terpenes, lectins) or natural delivery platforms (e.g., sulfated polysaccharides, chitosan), ideally in two separate subsections. Table 3, in its current form, does not add value and could be removed.
- L599-601 HSV is a dsDNA virus, and neither the nanovehicle nor the compound are natural. This sentence should be deleted.
- Figure 2 should be completed to reflect the administration challenges of all types of antiviral strategies involving natural compounds.
- Chapter 6, on databases, is interesting but disconnected from the central focus of the manuscript. Consider shortening or removing it if no concrete examples can be provided of how these databases have been used to discover, predict, or validate natural compounds. Alternatively, it could be integrated as a future perspective in the final discussion, in 1–2 paragraphs, rather than as a standalone chapter.
Regarding language and formal aspects, the English is generally understandable but contains stylistic and grammatical errors that hinder fluency. I recommend a full language revision.
The official nomenclature of the International Committee on Taxonomy of Viruses (ICTV) must be respected for taxonomic names, as well as the conventions for common virus names (https://ictv.global/report). Official taxonomic names (family, genus, species) should be capitalized and italicized. Common virus names (when referring to the virus as a pathogen) should be written in lowercase and without italics. For example:
- Common virus name: dengue virus (lowercase, non-italicized); the disease: dengue fever (lowercase, except at the beginning of a sentence); official species name: Orthoflavivirus denguei (italicized, with initial capital letter for Orthoflavivirus as per ICTV classification).
Example sentence: “The four serotypes of dengue virus, the etiological agent of dengue fever, belong to a single species (Orthoflavivirus denguei), which belongs to the genus Orthoflavivirus within the family Flaviviridae.”
Abbreviations should be defined at first mention and used consistently throughout. There is frequent alternation between full terms and abbreviations, as well as inconsistent use of italics for Latin terms; please unify.
Tables: align column contents to the left and top; the current centered formatting makes it difficult to determine where one item ends and the next begins.
Use the comma as the thousands separator in all cases.
Many of these remarks, along with additional specific comments, are highlighted in the annotated PDF of the manuscript.

Regarding language and formal aspects, the English is generally understandable but contains stylistic and grammatical errors that hinder fluency. I recommend a full language revision.
Author Response
Reviewer 1
The manuscript addresses a highly relevant and timely topic: the use of natural products in clinical and biomedical contexts for the treatment of RNA virus infections. The proposal is interesting and has great potential impact. However, I have identified several aspects that, in my opinion, should be revised to strengthen the clarity, focus, and coherence of the work. Below, I detail my observations with the intention of contributing to improving the quality and alignment of the manuscript with its stated objectives:
- The title promises a focus on RNA viruses and natural compounds reported in the last decade. However, in some passages, it is not entirely clear what the specific role of natural compounds is in relation to the “challenge” posed by RNA viruses.
Reply: We appreciate your comment. We have reworded the title to more accurately reflect the scope of the manuscript. The new title is:
“Natural Compounds with Antiviral Activity Against Clinically Relevant RNA Viruses: Advances of the Last Decade.”
This emphasizes that the review focuses on clinically meaningful RNA viruses, rather than all RNA viruses.
- The introduction mentions the use of biological matrices as bioreactors, but this topic is not revisited in the rest of the text. I suggest either developing it with concrete background examples or removing it.
Reply: We appreciate your comment. We have decided to remove the mention of biological matrices, such as bioreactors, from the introduction, as it was not included in the rest of the text. This allowed us to maintain the manuscript's coherence and primary focus.
- In the main body of the text, several sections provide very extensive descriptions of basic viral biology, sometimes in excessive detail (e.g., replication of influenza, rotavirus, Lassa). This may dilute the focus on natural antiviral compounds, which is what is expected according to the title and abstract. While the review is logically structured, there is an imbalance in length:
- The sections on RNA virus biology and basic replication mechanisms (chapters 2 and 3) are too extensive and detailed, with technical descriptions of well-known mechanisms that are not strictly necessary for the topic of the review. In my view, the information provided in Table 1 is sufficient. A caption should be added to Table 1 defining the abbreviations used for the viral types.
Reply: We appreciate your comment. We have condensed chapters 2 and 3, eliminating overly technical descriptions and retaining only the essential information. We have also added a legend to Table 1, clearly defining the abbreviations for the viral types.
- Chapter 4, dedicated to natural compounds, is shorter and more general than expected. It would be important to expand this section with concrete examples, mechanisms of action, results from animal models, and, if possible, clinical trial data. A critical comparison of results would significantly enrich the section. A much broader search of natural compounds with antiviral activity and the viral models in which they were tested is needed (for example, doi: 10.1002/cbdv.202401913; doi: 10.3390/nu12092534; doi: 10.1016/j.heliyon.2024.e27533).
Reply: We appreciate your feedback. Chapter 4 has been substantially expanded to incorporate specific examples of natural compounds, mechanisms of action, EC50 and CC50 data, as well as results from animal model studies and, where possible, clinical trials. In addition, recent references suggested by the reviewers have been incorporated, with a critical and comparative discussion of the results.
- L357: “The use of molecules obtained from organic matrices …” – what is meant by “organic matrices”? Should this not be “natural matrices”?
Reply: We appreciate your comment. The term "organic matrices" has been revised to "natural matrices" for greater precision and clarity.
- L370–376: Move the entire paragraph to section 4.2, Inhibition of host-cell entry. I recommend reorganizing Chapter 4 according to the different stages of the viral life cycle (entry, replication, release) to provide greater coherence.
Reply: We appreciate your comment. The highlighted paragraph was relocated to section 4.2 "Inhibition of host-cell entry," and Chapter 4 was reorganized according to the phases of the viral cycle (entry, replication, and release), which improves the coherence of the manuscript.
- Table 2 includes examples of synthetic rather than natural origin: celgosivir is a prodrug of a natural product (castanospermine) designed for oral administration and rapid conversion into its active form. UV-4B is a synthetic analog of natural DNJ, developed to improve its pharmacological and antiviral properties.
Reply: We appreciate your comment. The compound description in Table 2 has been revised to specify that celgosivir is a prodrug derived from castanospermine and that UV-4B is a synthetic analogue of DNJ. This clarifies their relationship to natural products.
- Section 4.4 should include only studies evaluating immunomodulatory effects in the context of viral infection, not those focusing on cancer, autoimmune diseases, non-viral infections, etc.
Reply: We appreciate your comment. Section 4.4 has been revised, retaining only studies on immunomodulation in the context of viral infections. References related to cancer, autoimmune diseases, and other non-viral conditions have been removed.
- The entire chapter should be reorganized and rewritten with consistency regarding which data will be provided for each example (CC50, EC50, in vitroor in vivo, biological source of the sample, type of extract, type of compound involved in the reported biological activity, etc.).
Reply: We appreciate your comment. To facilitate comparison among the different bioactive compounds reported, we summarized the main findings in Table 2, which compiles natural products with antiviral activity against RNA viruses described in the last decade. The table highlights the biological source, type of compound, experimental model, and key pharmacological values (CCâ‚…â‚€ and ECâ‚…â‚€) when available, thus providing a concise overview of their therapeutic potential.
- Chapter 5, focused on nanotechnology, is up to date but mostly describes applications related to synthetic antivirals. For a manuscript whose stated central focus is “natural antivirals,” this chapter appears weakly connected. I suggest emphasizing the link to natural compounds (e.g., flavonoids, terpenes, lectins) or natural delivery platforms (e.g., sulfated polysaccharides, chitosan), ideally in two separate subsections. Table 3, in its current form, does not add value and could be removed.
Reply: We appreciate your comment. We have reorganized Chapter 5 into two subsections to highlight its connection with natural compounds: (i) nanodelivery of natural antivirals (including flavonoids, terpenes, lectins, and peptides) and (ii) natural delivery platforms (such as chitosan, sulfated polysaccharides, and plant-derived vesicles). Furthermore, the contents of Table 3 were removed, and the information was integrated and synthesized into the text to improve the clarity and coherence of the manuscript.
- L599-601 HSV is a dsDNA virus, and neither the nanovehicle nor the compound are natural. This sentence should be deleted.
Reply: We appreciate your comment. The phrase mentioning HSV (a dsDNA virus) and its relationship to nanovehicles was removed, as it does not align with the manuscript's scope or focus on natural compounds.
- Figure 2 should be completed to reflect the administration challenges of all types of antiviral strategies involving natural compounds.
Reply: We appreciate your comment. We have redesigned Figure 2 to reflect the challenges of administering different types of natural antiviral compounds (peptides, flavonoids, terpenoids, polysaccharides, and lectins). Physicochemical, biological, and immunological barriers are now included, allowing for a broader view consistent with the central focus of the manuscript.
- Chapter 6, on databases, is interesting but disconnected from the central focus of the manuscript. Consider shortening or removing it if no concrete examples can be provided of how these databases have been used to discover, predict, or validate natural compounds. Alternatively, it could be integrated as a future perspective in the final discussion, in 1–2 paragraphs, rather than as a standalone chapter.
Reply: We appreciate your comment. Chapter 6 on databases has been reduced and integrated into the "Future Perspectives" section. It now appears in 1–2 paragraphs, illustrating how these tools support the discovery and prediction of natural compounds with antiviral potential.
- Regarding language and formal aspects, the English is generally understandable but contains stylistic and grammatical errors that hinder fluency. I recommend a full language revision.
Reply: We appreciate your comment. We have thoroughly reviewed the manuscript's language and formal aspects to correct grammatical and stylistic errors and to improve the text's fluency and clarity in English.
- The official nomenclature of the International Committee on Taxonomy of Viruses (ICTV) must be respected for taxonomic names, as well as the conventions for common virus names (https://ictv.global/report). Official taxonomic names (family, genus, species) should be capitalized and italicized. Common virus names (when referring to the virus as a pathogen) should be written in lowercase and without italics. For example:
- Common virus name: dengue virus (lowercase, non-italicized); the disease: dengue fever (lowercase, except at the beginning of a sentence); official species name: Orthoflavivirus denguei(italicized, with initial capital letter for Orthoflavivirus as per ICTV classification).
Example sentence: “The four serotypes of dengue virus, the etiological agent of dengue fever, belong to a single species (Orthoflavivirus denguei), which belongs to the genus Orthoflavivirus within the family Flaviviridae.”
Reply: We appreciate your comment. The viral nomenclature was reviewed and corrected in accordance with ICTV guidelines. Official taxonomic names were capitalized and italicized, and common virus names were kept in lowercase and not italicized.
- Abbreviations should be defined at first mention and used consistently throughout. There is frequent alternation between full terms and abbreviations, as well as inconsistent use of italics for Latin terms; please unify.
Reply: We appreciate your comment. All abbreviations were defined upon first mention, and their use was standardized throughout the manuscript. The consistency of the use of Latin terms in italics was also corrected.
- Tables: align column contents to the left and top; the current centered formatting makes it difficult to determine where one item ends and the next begins.
Reply: We appreciate your comment. The tables have been adjusted.
- Use the comma as the thousands separator in all cases.
- Many of these remarks, along with additional specific comments, are highlighted in the annotated PDF of the manuscript.
Reply: We appreciate your comment. We have carefully reviewed the annotated PDF and made the corrections noted in the manuscript.
Reviewer 2 Report
Comments and Suggestions for Authors
The manuscript by David Mauricio Cañedo-Figueroa et al. focuses on discussing the recent advances in the identification of bioactive compounds that exhibit inhibitory activity against clinically relevant viruses, including the influenza virus, HIV, dengue virus, Zika virus, and SARS-CoV-2. The manuscript is based on the review of many sources and will probably be useful for researchers as a systematization of knowledge in this area. However, in its current state, the manuscript requires revision.
- It is unclear how the authors selected papers for the review, what the selection was based on, my proposal to include a small section on materials and methods to explain the basis for selecting papers, search words, databases, etc.
- In this regard, the next question is, in fact, there are many more RNA viruses, as follows, for example, from this paper (https://www.sciencedirect.com/topics/neuroscience/rna-virus), many of them are pathogens for humans. However, for some reason, the authors selected only a few viruses. In this case, the title may be misleading to readers. I recommend rephrase the title, since it is currently very broad and mean all RNA viruses. I also recommend writing a criterion for selecting viruses for the review.
- The review contains some possibly irrelevant sections that can be shortened or omitted. In particular, the text on antiviral peptides is unclear how relevant it is to the topic of bioactive flavonoids, alkaloids, terpenes, lectins, and polysaccharides. There is not a word about peptides in the abstract. Either change the abstract or remove this section from the text.
Table 3. This table needs to be better structured, since It is currently overloaded with text in column 2 and not easy to understand. Consider significantly reducing the text in this column, and moving most of the text to the body of the manuscript.
Author Response
Reviewer 2
The manuscript by David Mauricio Cañedo-Figueroa et al. focuses on discussing the recent advances in the identification of bioactive compounds that exhibit inhibitory activity against clinically relevant viruses, including the influenza virus, HIV, dengue virus, Zika virus, and SARS-CoV-2. The manuscript is based on the review of many sources and will probably be useful for researchers as a systematization of knowledge in this area. However, in its current state, the manuscript requires revision.
- It is unclear how the authors selected papers for the review, what the selection was based on, my proposal to include a small section on materials and methods to explain the basis for selecting papers, search words, databases, etc.
Reply: We sincerely thank the reviewer for this valuable suggestion. Following the recommendation, we have added a new Section 4. Materials and Methods, where we describe the databases consulted (PubMed, Scopus, Web of Science), the period covered (2013–2024), the keywords used, and the inclusion/exclusion criteria applied to select the studies considered in this review.
- In this regard, the next question is, in fact, there are many more RNA viruses, as follows, for example, from this paper (https://www.sciencedirect.com/topics/neuroscience/rna-virus), many of them are pathogens for humans. However, for some reason, the authors selected only a few viruses. In this case, the title may be misleading to readers. I recommend rephrase the title, since it is currently very broad and mean all RNA viruses. I also recommend writing a criterion for selecting viruses for the review.
Reply: We appreciate your comment. We have reworded the title to more accurately reflect the scope of the manuscript. The new title is:
“Natural Compounds with Antiviral Activity Against Clinically Relevant RNA Viruses: Advances of the Last Decade.”
This emphasizes that the review focuses on clinically meaningful RNA viruses, rather than all RNA viruses. We have also added the criteria used to select viruses included in the review to the Materials and Methods section.
- The review contains some possibly irrelevant sections that can be shortened or omitted. In particular, the text on antiviral peptides is unclear how relevant it is to the topic of bioactive flavonoids, alkaloids, terpenes, lectins, and polysaccharides. There is not a word about peptides in the abstract. Either change the abstract or remove this section from the text.
Reply: We appreciate your comment. We have adjusted the manuscript to more clearly integrate the relevance of antiviral peptides within the framework of natural compounds. The introduction and Chapter 5 have been updated to explain their role as part of nanodelivery strategies and biocompatible platforms. Furthermore, the abstract has been modified to include explicit mention of peptides, ensuring consistency between the content and the manuscript's overall focus.
Table 3. This table needs to be better structured, since It is currently overloaded with text in column 2 and not easy to understand. Consider significantly reducing the text in this column, and moving most of the text to the body of the manuscript.
Reply: We appreciate your comment. We have removed Table 3 and summarized the information in the body of the text.
Round 2
Reviewer 1 Report
Comments and Suggestions for Authors
The manuscript still presents several writing errors that need to be addressed and corrected.

Author Response
We sincerely appreciate the reviewer’s observation. The entire manuscript has undergone a thorough language revision to correct grammatical, syntactic, and stylistic issues.
Reviewer 2 Report
Comments and Suggestions for Authors
The authors have provided responses to all my questions therefore I do not have any further comments
Author Response
Thank you very much for your time in reviewing.